# Menstrual Hygiene Preparedness Among Schools in India: A Systematic Review and Meta-Analysis of System-and Policy-Level Actions

**DOI:** 10.3390/ijerph17020647

**Published:** 2020-01-19

**Authors:** Shantanu Sharma, Devika Mehra, Nele Brusselaers, Sunil Mehra

**Affiliations:** 1Researcher, Department of Clinical Sciences, Lund University, 22100 Lund, Sweden; 2MAMTA Health Institute for Mother and Child, Delhi 110048, India; dr_mehra@mamtahimc.org; 3Public Health Consultant, Medeon Science Park, 21432 Malmö, Sweden; 21.devika@gmail.com; 4Centre for Translational Microbiome Research, Department of Microbiology, Tumor and Cell biology, Karolinska Institute, SciLifeLab, 17177 Stockholm (Solna), Sweden; nele.brusselaers@ki.se

**Keywords:** Adolescent, Education, Health, Hygiene, Sanitation

## Abstract

With increasing girls’ enrolment in schools, school preparedness to ensure a menstrual friendly environment is crucial. The study aimed to conduct a systematic review regarding the existing evidence on menstrual hygiene management (MHM) across schools in India. It further aimed to highlight the actions that have been taken by the government to improve the MHM situation in India. We conducted the systematic literature search using PubMed, EMBASE, and Web of Science for searching the peer-reviewed articles and Google Scholar for anecdotal reports published from inception until 30 October 2019. Of 1125 publications retrieved through the search, 183 papers were included in this review, using *a priori* created data-extraction form. Meta-analysis was used to estimate the pooled prevalence (PP) of MHM practices in schools. Less than half of the girls were aware of menstruation before menarche (PP 0.45, 0.39 to 0.51, *I*^2^ = 100.0%, n = 122). Teachers were a less common source of information about menstruation to girls (PP 0.07, 0.05 to 0.08, *I*^2^ = 100.0%, n = 86). Separate toilets for girls were present in around half of the schools (PP 0.56, 0.42 to 0.75, *I*^2^ 100.0%, n = 11). MHM in schools should be strengthened with convergence between various departments for explicit implementation of guidelines.

## 1. Introduction

Inadequate menstrual hygiene management (MHM) among adolescent girls (15–19 years) is a public health problem, mainly in low and middle-income countries [1]. With over 0.6 billion adolescent girls (8% of the world’s population), the issue of menstrual hygiene by virtue of its magnitude is an issue of global concern. More than 80 percent of these adolescents reside in the Asian and African continents [2]. India is home to 243 million adolescents, which accounts for a quarter of the country’s total population [3]. India has over 355 million menstruating women and girls, but millions of women across the country face uncomfortable and undignified experience with MHM [4].

Menstrual hygiene is often regarded as a multi-sectoral issue that requires an integrated action from the Department of Education, Health, Women, and Child Development and Water Sanitation Hygiene (WASH) [5]. In recent years, we have witnessed a strengthened move by the government towards addressing this public health issue. With the launch of the National Rural Health Mission in 2005, menstrual hygiene promotion was formally included as a key responsibility of the community health workers (Accredited Social Health Activist; ASHA) followed by the implementation of menstrual hygiene promotion scheme for girls in rural areas in 2011 [6]. In 2015, another milestone was achieved when the Ministry of Drinking Water and Sanitation published guidelines on MHM [7]. There has been a lot of national and international level push to address this issue through various social media platforms including the making of a film called Padman [8], roll out of menstrual hygiene campaigns, performing trials on eco-friendly or biodegradable menstrual products, implementing comprehensive sexuality education in schools, etc. [4].

Menstruation among school-age girls is a neglected issue on the implementation front despite the formal inclusion of a menstrual hygiene scheme under the reproductive and child health program by the government of India (in 2011) [9]. This issue still lacks educational support from health workers, pragmatic guidelines to operationalize MHM in schools, and adequate monetary resources to implement the needed actions. Fear, shame, ongoing social taboos, ignorant unsupportive teachers, lack of water, sanitation, disposal facilities, and privacy, are some of the barriers in building an enabling environment for safe and hygienic menstrual practices within the school premises [10,11,12]. These system-level challenges, in conjunction, not only negatively impact sexual and reproductive health outcomes of adolescent girls but also affects their self-confidence and agency (ability to make a decision and take actions for self) [11]. The increasing enrolment of girls in secondary and senior secondary schools demands a more comprehensive approach to make schools menstrual hygiene friendly and prevent school dropouts or absenteeism [13].

Given that a comprehensive approach to study MHM among schools in India was not made in previous reviews, we chose to conduct a systematic review. The review aimed to objectively summarize the evidence on the actions taken at the school (system)- and policy-level to make schools a menstrual hygiene friendly place for adolescent girls in India.

### Research Question

The research question was defined as “Are schools in India menstrual hygiene friendly, and what are the policy-level actions taken by the government of India to make our schools menstrual hygiene friendly?”

## 2. Materials and Methods 

### 2.1. Literature Search

We used the PRISMA (Preferred Reporting Items for Systematic Reviews and Meta-Analyses) framework for systematic reviews to identify the published and grey literature on school (system-) and policy-level actions [14]. A systematic literature search was undertaken to identify the evidence using the online databases of MEDLINE (PubMed), EMBASE, and Web of Science, from inception until 30 October 2019. In addition, for the policy-level actions, we did a manual search for the relevant documents through the government of India’s website and Google Scholar. We searched the websites of the four concerned ministries related to MHM, including the Ministry of Health and Family Welfare, Ministry of Women and Child Development, Ministry of Drinking Water and Sanitation, and Ministry of Human Resource Development. Since many papers, project reports, documents, or guidelines are not published in peer-reviewed journals, the Google Scholar search was extended to include the grey literature.

Keywords used for the search across three databases were: (‘Menstruation OR Menstrual OR Menses OR Periods OR Hygien* OR Sanitation OR Sanitary OR Hygiene’) AND (‘School OR Adolescent OR Adolescen* OR Pubescence’ AND ‘Girl OR Women OR female’) AND India. Cross-referencing (screening reference list of included studies) was also used to add other studies of relevance to our review. We did not consider abstracts from conference books, manuscripts, or reports published in any language other than English. Endnote X.8.0.1 (Clarivate Analytics, Philadelphia, US) was used to manage all references identified in the search. All the search results were imported in endnote and duplicates if any were removed.

### 2.2. Inclusion and Exclusion Criteria

For the purpose of this review, a ‘menstrual hygiene friendly school’ was defined as schools where (1) teachers had adequate knowledge about MHM or teacher was a source of information for MHM (before or after menarche) to girls, (2) school management committees took menstrual health-promoting actions, and (3) there were facilities of clean, separate girls’ toilets, changing rooms, water, soap, safe disposal of used pads, and emergency sanitary pads (sanitation facilities). Furthermore, (4) male sensitization on MHM, (5) girls’ awareness on menstruation before menarche, (6) availability of education material on menstrual hygiene promotion, (7) waste management facilities in school premises, and (8) regular monitoring of the schools for rolling out MHM, were added dimensions of menstrual hygiene friendly schools [12]. We established inclusion criteria as any publication that described any of the above-said components of menstrual hygiene friendly schools in India for school-level actions (Figure 1). Any study not reporting on any of these eight dimensions was excluded during the screening process. Studies that included girls’ awareness of MHM in their findings but did not mention that they were school-going girls were also excluded from the analysis. For policy-level actions, guidelines, or reports that have specified about the government of India’s actions on MHM promotion in schools were included. 

### 2.3. Data Analysis

Quality assessment of included studies was performed based on seven criteria as specified in another review [15]. Each criterion had a value of one or zero. For each study, the results of all the seven criteria were summed to obtain a quality score ranging from 0 to 7. However, studies were not excluded on the basis of a quality score. The quality assessment sheet of the includes studies is provided as Appendix A. We did not contact the authors of the studies or reports for further information. Two authors (Shantanu Sharma and Devika Mehra) independently reviewed all the titles and abstracts to select the relevant studies. The data on the above-said dimensions, as specified previously, were extracted from the included studies according to a standard form created *a priori*. Discordance between the two authors was resolved by consensus. The results are presented based on the eight components of menstrual hygiene friendly schools. 

Meta-analysis was performed on four out of the eight components of school-level actions as quantitative data were available for only four of them. These four components included teacher as a source of information about MHM (before or after menarche) to girls, separate toilet facilities for girls in schools, awareness of girls on menstruation before menarche, and good disposal facility for sanitary pads in schools. Pooled prevalence (PP) was estimated in a random-effects model using the RevMan version.5.3 software (Cochrane Collaboration, London, UK). Forest plots were generated to display the overall random-effects pooled estimates with 95% confidence intervals. The heterogeneity was quantified using the *I*^2^ measure and its confidence interval. We used generic inverse variance method and computation of the standard error was done using the formula: Square Root [(proportion*(1-proportion))/sample size]. 

## 3. Results

Of 1125 papers and reports (of which 152 retrieved through cross-referencing and grey literature), 183 were considered eligible (Figure 2). Furthermore, 153 out of 183 were used for the quantitative synthesis of meta-analysis. Most of the studies were of low-to-moderate quality with the combined average score of 2.6. The characteristics of the included papers have been shown in Table 1. The PRISMA framework checklist is provided as Appendix A. The findings from included papers were presented under two broad themes, namely, school-level (system) actions, and policy-level actions. Under the system-level (school) actions, eight components were included, as defined previously.

### 3.1. School-Level Actions (System)

#### 3.1.1. Menstrual Hygiene Knowledge Among School Teachers

According to our literature review, there was no peer-reviewed publication or anecdotal reports on the knowledge of school teachers regarding menstruation issues in India. The programs that included reproductive health education as a means to disseminate MHM information in schools did not measure or mention this. Moreover, school teachers were reported as the less common source of MHM information among adolescent girls in 74 studies [16,17,18,19,20,21,22,23,24,25,26,27,28,29,30,31,32,33,34,35,36,37,38,39,40,41,42,43,44,45,46,47,48,49,50,51,52,53,54,55,56,57,58,59,60,61,62,63,64,65,66,67,68,69,70,71,72,73,74,75,76,77,78,79,80,81,82,83,84,85,86,87,88,89], yet in 12 studies, a large proportion of girls (more than one-fourth) reported that teachers were a common source of information about MHM (Table 2) [90,91,92,93,94,95,96,97,98,99,100,101]. The teachers in those schools were reported to have been supportive [95]. In addition to these challenges, non-availability or limited availability of female teachers in schools was a serious issue. Despite schools running health programs, teachers found discussing menstruation embarrassing and instruct the students to read that chapter in the textbook at home. As English was not taught in some schools, the use of vernacular terms for human reproductive organs in the local language became very embarrassing for teachers as well as students [102]. It has been reported that many teachers were insensitive to the physical and mental state of girls during their periods [16]. Moreover, a study from Tamil Nadu reported that 1.3% of the schoolgirls were scolded by teachers for menstrual problems [103]. Teachers felt the need to do games and activities to share information about various menstruation-related issues with girls. However, they could not conduct such activities in school because they were pre-occupied with routine duties and curriculum targets [77]. Teachers were a source of information among 7% girls (PP 7.0%, 95% Confidence Interval (CI) 5.0% to 8.0%, *I*^2^ = 100%, n = 86) (Figure 3).

#### 3.1.2. School Management Committee Taking Menstrual Health-Promoting Actions 

The World Health Organization proposed the concept of Health Promoting Schools (HPS) in 1995, which advocated for the total life approach to school-based health promotion while focusing on the curriculum, the school’s ethos, and the environment. The HPS framework emphasized creating a management committee as a support structure for schools to help in planning, designing policies, strategies, and procedures towards health promotion [104]. Menstrual hygiene promotion could be one of the outcomes of the actions of this committee. However, evidence on the existence of such committees and their commitment to health promotion was limited [105]. In two intervention studies from Bihar and Chandigarh, the school management committee under the HPS framework was established as an effective means towards promoting health [105,106]. School management committees were non-functional and completely unaware of their roles and responsibilities [107,108].

#### 3.1.3. Sanitation Facilities in Schools

Unavailability of disposal mechanisms for pads, poor water supply for washing or flushing, poor hygienic conditions of the toilets, lack of soap, washbasins, mugs for washing in the toilets, and no separate toilets for girls were major WASH challenges girls faced during menstruation. Broken lock/doors of the toilets were a matter of concern for the security of the girls in schools. These findings have been reported in 30 studies [26,33,72,77,78,95,102,105,106,109,110,111,112,113,114,115,116,117,118,119,120,121,122,123,124,125,126,127,128,129]. There were gaps with respect to the non-availability of emergency supplies of sanitary materials in schools [26,69,72,78,95]. The girls in schools threw away sanitary pads or other menstrual articles in toilets or left the soiled wrapped pads at toilet corners due to lack of dustbins or separate place for disposal. As a result, the sewage system was blocked, or toilets became dirty, a breeding place for flies and mosquitoes, and unhygienic for other toilet users and cleaners [26,77,109]. Less than two-thirds of schools in India had dustbins with a lid for disposing of pads. The proportion of schools with bins having lids for the disposal of sanitary materials was 62%. [110]. Moreover, only 21% of girls could get pain relievers for menstrual cramps, and 37% told that absorbents were available in schools when needed [78]. Only 56% (PP 56%) schools in India had separate toilets for girls (95% CI 42% to 75%, *I*^2^ = 100%, n = 11) (Figure 4). Table 3 shows the characteristics of some of the studies that reported on the presence of separate toilets for girls in schools. 

#### 3.1.4. Sensitization of Boys and Male Teachers on MHM

In 2014, UNESCO, in its technical note, emphasized that male teachers in the schools might not be sensitized to the needs of girls, and hence, did not allow them to visit the toilet during their lecture. Male teachers perceived that girls were not interested in studies [130]. In other studies, it was reported that teasing by male teachers was common. This insensitive behavior might be fueled by ignorance, prevailing local myths, and cultural taboos related to menstrual blood among men [130,131]. As a result, topics such as puberty and menstruation were not included in the curriculum due to the predominance of male teachers in most of the schools [16,77]. There was a higher predominance of male teachers or administrators in schools, and they were hesitant to talk about MHM due to the gendered rooting of menstruation and cultural taboos related to it. It was reported that girls were often teased and subjected to embarrassment by boys and male teachers in schools due to the staining of their clothes during periods [130]. Moreover, because of the lack of knowledge about menstruation, boys displayed a negative attitude towards menstruation [132].

#### 3.1.5. Girls’ Awareness of MHM

Another barrier to a comfortable and dignified experience of MHM among girls was the lack of or limited awareness as reported in 92 studies (Table 2) [16,18,23,24,25,26,31,33,34,35,36,37,39,40,41,42,43,44,48,49,51,52,53,54,55,57,58,62,63,64,65,67,68,69,70,71,77,78,80,81,82,83,84,85,87,89,90,92,93,95,97,99,133,134,135,136,137,138,139,140,141,142,143,144,145,146,147,148,149,150,151,152,153,154,155,156,157,158,159,160,161,162,163,164,165,166,167,168,169,170,171,172]. The lack of awareness led girls to think menstruation as a representation of sin, and menstrual blood as an impure entity. Schools were not reported often as a source of menstrual hygiene education [78]. On the contrary, 34 studies documented that a large proportion of girls (more than two-third) had high knowledge about menstruation [19,22,28,29,30,45,47,60,76,79,88,94,96,98,100,173,174,175,176,177,178,179,180,181,182,183,184,185,186,187,188,189,190,191]. Among 122 studies with available information, the pooled prevalence of awareness about menstruation before menarche was 45% (95% CI 39% to 51%, *I*^2^ = 100%) (Figure 5). There was not much difference in the proportion of girls who were aware about menstruation before menarche between rural and urban areas (around 2%, not shown in the data).

#### 3.1.6. Education Material for Menstrual Hygiene Promotion in Schools

Limited data were available related to this component of MHM in schools. The study among schools across three states of India (Chhattisgarh, Maharashtra, and Tamil Nadu) reported that written materials about menstruation were infrequently available (19%) in schools [78]. In the global baseline report 2018, it was reported that around 64% of schools in India were providing menstrual hygiene education to female students [110].

#### 3.1.7. Facilities for Waste Management in Schools

Safe waste management of used sanitary pads in schools was another major issue. Most of the schools lacked any such facility. The lack of facilities discouraged girls from using sanitary pads in schools or attending schools during menstruation [72,75,78,102,109,123]. School sanitation and hygiene education under the total sanitation campaign in Uttar Pradesh, the largest state in India, had the provision of installing incinerators in toilets of secondary schools, but none of the schools implemented the same [114]. It was reported in a study that only 27% of schools had good disposal facilities for menstrual waste on their premises (Table 3). The most frequently mentioned option for disposal was taking the soiled item home (21%) followed by burn pits (20%), rubbish pits (17%), bins (16%), and incinerator (7%). Incinerators were common in selected states and certain grades of schools [78]. It has been reported that some schools used incinerators or “feminine hygiene bins” for disposing menstrual waste material, but due to shyness or fear of being seen by others, they were not used. Sanitary napkin vending machines have been installed in toilets of some schools in Kerala, which are semiautomatic and operated by inserting a coin in it. It contained 30–50 sanitary napkins to meet the emergency needs of the girls/women in schools [109]. In the joint WHO-UNICEF baseline report 2018, it was reported that only 36% of schools in India had functional incinerators for disposal of sanitary wastes. Mizoram is the only state where more than 50% of schools have a functional incinerator for the disposal of sanitary waste. [110]. Thirty percent of the schools had good disposal facilities for sanitary products (PP 30%, 95% CI 13% to 69%, *I*^2^ = 100%, n = 2) (Figure 6). 

#### 3.1.8. Monitoring of Schools for MHM Friendly Services

The concept of periodic monitoring of the data related to MHM practices in schools was in its nascence, and a key focus on measuring outcome indicators needs to be levied. To our knowledge, there was no school-based data on such measures.

##### Policy-Level Actions to Address MHM in Schools

The first in the series of national-level directions on MHM for schools was the operational guidelines for the promotion of menstrual hygiene (2012) in rural areas [9]. The guidelines outlined the strategy to reach school girls through the adolescent education program. The key components of the school-based program were the provision of sanitary napkin distribution, health education, and incinerator for safe disposal. In 2014, the Ministry of health and family program launched the National Adolescent Health Programme known as *Rashtriya Kishore Swasthya Karyakaram* (RKSK), which levied clear guidelines for providing education, awareness, and support for better MHM using the peer education model. This national program worked at building protective factors that could help adolescents developing ‘resilience’ through both community and school-based interventions [192]. In the recent five years, sanitation and hygiene received a much-needed impetus from stakeholders of all spheres. With the launch of menstrual hygiene management guidelines in 2015 [7], the issue was streamlined into a formal agenda. The action guide laid down the suggestive measures to ensure menstrual hygiene friendly schools. The guidelines addressed the performance measurement with six indicators dedicated to assessing school performance based on MHM. However, there was a lack of detailing on the process and activity-oriented charting of the MHM framework, which schools would follow.

Another milestone in this realm towards filling MHM gaps in schools was a comprehensive WASH assessment tool. It was operationalized in a three-year project led by the Urban Management center and supported by the government of Gujarat [193]. The tool underscored the need for MHM facilities and IEC across schools besides key components of WASH infrastructure assessment. The initiative (2014–2017) envisaged innovative approaches such as mobile application-based data collection for school sanitation surveys, competition-based approach to WASH improvement named as school *swachh survekshan* (cleaniness assessment), the concept of creating ‘model school’ based on Indian standard codes and *Sarva Shiksha Abhiyan* (a program for universal elementary education) standards with a positive environment for integrated learning, sports, recreation, and good access to WASH facilities. This joint action research program involved behavior change approaches such as IEC campaigns, school sanitation clubs, self-assessment tools for monitoring sanitation index, etc. [193].

The push for MHM at an international level contributed towards sailing the agenda across nations, including India. Understanding the importance and growing interest in transforming the school environment for menstruating girls and female teachers, the “MHM in ten” members put forward a 10-year agenda (2014–2024). The five key action priorities of the plan revolved around building a strong cross-sector evidence base for MHM in schools, around developing and disseminate guidelines, do evidence-based advocacy, delegate responsibility, and integration with the education system [194]. The recently released Clean India: Clean Schools handbook underpinned the theme of securing a healthy school environment [195]. Installation of the napkin-vending machines and environmentally safe disposal mechanisms such as low-cost incinerators attached to the girls’ toilets in schools for disposal of used MHM products were major efforts in this direction. The government launched 100 percent oxy-biodegradable sanitary napkins under the name “*Suvidha* (facility)”. These sanitary pads were available under the scheme, entitled “*Pradhan Mantri Bhartiya Janaushadhi Pariyojana*” (Prime Minister Indian People Drug Scheme). The sanitary pads were made available at INR 1 in the drug dispensing stores created under the scheme. These napkins biodegrade automatically when it comes in contact with oxygen after being discarded [196].

## 4. Discussion

Menstrual health promotion in schools remains an issue of concern in India. Limited evidence was available on the different components of menstrual hygiene friendly school. Most of the evidence was available on two components, primarily girls’ awareness about MHM, and sanitation facilities in schools, leaving other components unaddressed. MHM in schools, although it was conceptualized comprehensively with different components as documented in guidelines, the data on its implementation was limited. There was a dearth of literature on education programs focusing on MHM in schools and knowledge, attitude, practices of mentors (teachers) who acted as an immediate source of information to girls. Although the data were available for the source of information about MHM (teachers), the studies on whether teachers as a source of information to girls had adequate knowledge about MHM were not available. We estimated that more than half of the girls did not have information about menstruation prior to menarche. Only 7% of girls reported teachers as a source of information for MHM. Menstruation hygiene education in school has most often being outsourced to non-governmental agencies [197,198]. Discrimination against female teachers to continue teaching in schools during periods was another example of a social barrier against menstruation. Not only did this practice disrupt the learning process, but it also perpetuated negative images among young minds and society [199].

Research evidence revealed that lack of sanitation facilities in schools hindered the ability of girls to manage menstruation healthily, safely, and with dignity. Evidence showed how this aspect affected coping strategies of girls during menstruation [5]. Only 56% of schools had the facility of a separate toilet for girls. Appropriate menstrual waste disposal facilities were still lacking in the majority of the schools in the country. Studies reported that because of a lack of awareness and sanitation facilities, most of the girls did not change pads in schools [23,78,111]. Despite being emphasized in the education policies, display of MHM messages through information, education, and communication (IEC) materials were not routinely practiced in schools [200]. IEC materials such as posters, leaflets helped to reinforce the health promotion messages and supporting behavior change at large [201]. Although online monitoring of some of the WASH indicators in schools was done, MHM components were not included [202]. A lack of evidence on MHM management information system (MIS) data takes away the system of their efficacy in dealing with this social health problem at a large scale [203]. Other reviews have reported similar findings on one or more of the eight components of MHM friendly schools in India [15,204,205]. 

Our review highlighted minimal rural-urban differences in menstrual hygiene practices in schools. However, in the national-level survey, it was reported that more than 50% of the rural girls did not use hygienic methods of menstrual protection (girls who use locally prepared napkins, sanitary napkins, or tampons during their menstrual period) compared to 23% in urban areas [206]. The plausible explanation for this could be the heterogeneity in the included studies in our review. Furthermore, most of the studies had low quality scores. 

It is imperative to emphasize the four primary considerations to build effective evidence on MHM friendly school aspect. These are discussed further, below.

Firstly, pre-service training of teachers on MHM with knowledge assessment at regular intervals is a crucial step in this regard since teachers are viewed as health promoters [195,207]. Teachers’ knowledge assessment can be a part of the regular school education surveys [13]. Furthermore, the sensitization of male teachers and boys on MHM is equally important. The provision of MHM-related education materials in schools such as booklets, flipcharts, and modules can be the cornerstone in enhancing the knowledge of teachers and girls [208].

The second major issue is the urgent need for improvement in the sanitary facilities at schools. MHM was missing in the majority of the schools [194]. We found data that highlighted the poor sanitation facilities across the school, and effective implementation and monitoring on this aspect were awaited. Previously published meta-synthesis highlighted that the poorly supportive physical infrastructure, such as a lack of water and sanitation facilities, made it difficult for girls to practice MHM safely [208]. Waste disposal is of equal concern to make the school environment clean and healthy. The widespread reality of poor sanitary facilities and ignorance about menstruating girls’ needs in schools can make its experience a negative one resulting in increased dropout rates among girls [209]. 

The third major area is the efficient working of the school management committee with an emphasis on MHM services in schools. Regular monitoring and timely actions are crucial to transform poor MHM practices in schools. Lastly, an efficient MIS is paramount in constructing evidence-based planning for the policymakers and the education leaders. Improved management of supplies and data generation demands an MIS software to update school authorities and concerned departments in the government at regular intervals. The MIS software may generate monthly data regarding the menstrual supplies stock, availability of sanitation facilities across schools, count of the menstruating adolescent girls, and school preparedness towards maintaining sanitation friendly status [210]. The Education MIS under UNICEF’s WASH programs (Wins) in schools across 194 countries provides a classic example of robustness and usefulness of data monitoring [211].

The multi-sectoral approach to MHM gaps in schools calls for convergence among various Departments such as Health and Family Welfare, Human Resource Development, Tribal Affairs, Woman and Child Development beyond the Department of Drinking Water and Sanitation. We need to leverage the use of resources and concentrated efforts to support school-based interventions for MHM. The different components to make schools menstrual hygiene friendly have been prioritized in other resources [212].

The ad-hoc grant-based projects or pilot initiatives by external agencies on MHM in schools are essential for evidence generation, which can be scaled-up as cost-effectiveness solutions at the national or state level. WaterAid India and Vatsalya (Breaking the Silence program) in Uttar Pradesh were working with the schools and service providers to change the perception around MHM [213]. The program proactively engaged with boys, school teachers, and management committees. Another intervention called ‘the MHM curriculum’, implemented by WASH United India, adopted game-based approaches across schools to empower girls in overcoming the stigma around menstruation [214]. Under the broad school health-promoting framework, knowledge and perceptions around menstruation were addressed with the support of lay counsellors in the SEHER (Strengthening Evidence base on scHool-based intErventions for pRomoting adolescent health) project from Bihar. This randomized control trial advocated for the involvement of lay counsellors in transforming the school climate and improving adolescent health outcomes [105].

Multiple non-peer reviewed anecdotal evaluation reports and articles documented that the implementation of such school-based MHM interventions was imperative to construct evidence. One such evidence was from a large-scale study covering 15 districts in India, called project *JAGRITI*, with menstrual hygiene promotion among adolescent girls as one of the components [215]. The program, run by the MAMTA-health Institute for Mother and Child, made a 10-step pragmatic guideline towards transforming schools into menstrual hygiene friendly with essential and desirable components (adapted from the National guidelines). Other national and state-level menstrual health players active in India are contributing to the availability of low-cost disposable sanitary material, MHM education to girls through comic books, training of facilitators, and researching on MHM behavior and practices [4].

Poor menstrual hygiene practices can lead to potential long-term consequences such as dropping out of school, early marriage, restriction of mobility, agency development (capacity to act independently), menstrual irregularities, and other reproductive and mental health problems. Moreover, menstrual irregularities during reproductive age group are common in many gynecological diseases, such as endometriosis, which may affect mental and psychological well-being in long-term [216]. There are multiple challenges girls face in managing menstruation due to poor awareness about safe practices, limited access to sanitary products, sanitation, and lack of support from teachers or family members. Schools have emerged as an important delivery platform for health promotion interventions, which needs more consistent efforts to improve the health outcomes of young girls.

### Limitations of the Study

The results of the review should be interpreted in view of some limitations. This review aimed to provide an overview of menstrual hygiene practices in schools. We could not produce a critically appraised and synthesized results for all the components of menstrual hygiene friendly schools. Heterogeneity between the included studies was very high, which might affect the validity of the pooled results. Most of the included studies were of low quality. The reports and peer-reviewed journal articles, which were publicly available, were included in our study. This limits our access to published literature in the public domain only. The study results might be considered in lieu of publication bias for positive findings because negative findings might not have been placed in the included reports and papers or papers and reports with negative findings may not have been published or made publicly available. 

## 5. Conclusions

MHM practices in schools are poor in India. Furthermore, we lack sufficient data to conclude the MHM situation in schools. The government has developed national level guidelines on all the aspects of MHM friendly school. However, its effective implementation on the ground is lacking. Still, MHM in schools is largely supported by outside agencies. Research on MHM in schools is mainly focused on observational studies to assess the knowledge and practices of girls regarding MHM. Moreover, research on the other aspects, such as waste management, teacher’s knowledge assessment, and management information, is limited.

There is a wide scope of integrating various curriculum or non-curriculum-based actions on menstrual health education and establish schools as an ideal forum to disseminate MHM information. There is a need for transforming the existing infrastructure into menstrual hygiene friendly, which needs to be the priority area for all the schools (government or private). Simplifying the elaborated guidelines into pragmatic action points would help authorities and management committees to implement the program easily in all the schools. The increased momentum from international donors, small and medium-sized enterprises, and non-governmental organizations could be synergized and channeled into constructive outcomes for attaining improved menstrual health outcomes. The emerging scientific and innovative solutions from MHM projects could help policymakers in strategizing concentrated efforts in this direction. Moreover, expanding MHM accountability from sanitation and health ministries to other departments will help to improve menstrual hygiene conditions in the country multilaterally. To better understand the problems surrounding MHM for adolescent girls in school, the impact of MHM interventions, we need new research studies with expanded range of methodologies.

## Figures and Tables

**Figure 1 ijerph-17-00647-f001:**
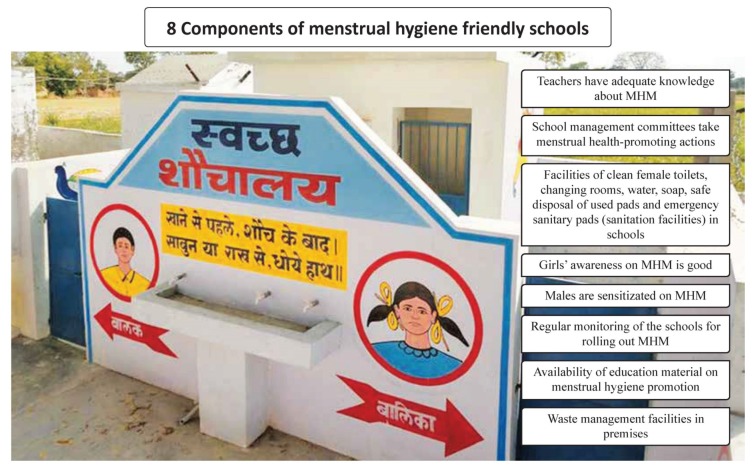
The figure depicts the photo of a well-designed toilet infrastructure in one of the States in India. Moreover, the figure depicts the eight components of menstrual hygiene friendly schools, which were used as the inclusion criteria for studies in this review. (MHM: Menstrual Hygiene Management).

**Figure 2 ijerph-17-00647-f002:**
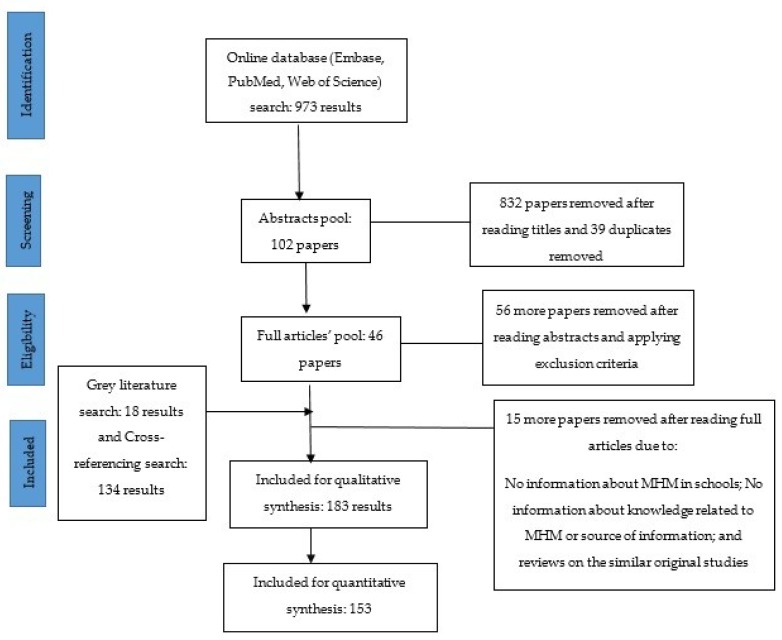
Search and exclusion criteria for literature review.

**Figure 3 ijerph-17-00647-f003:**
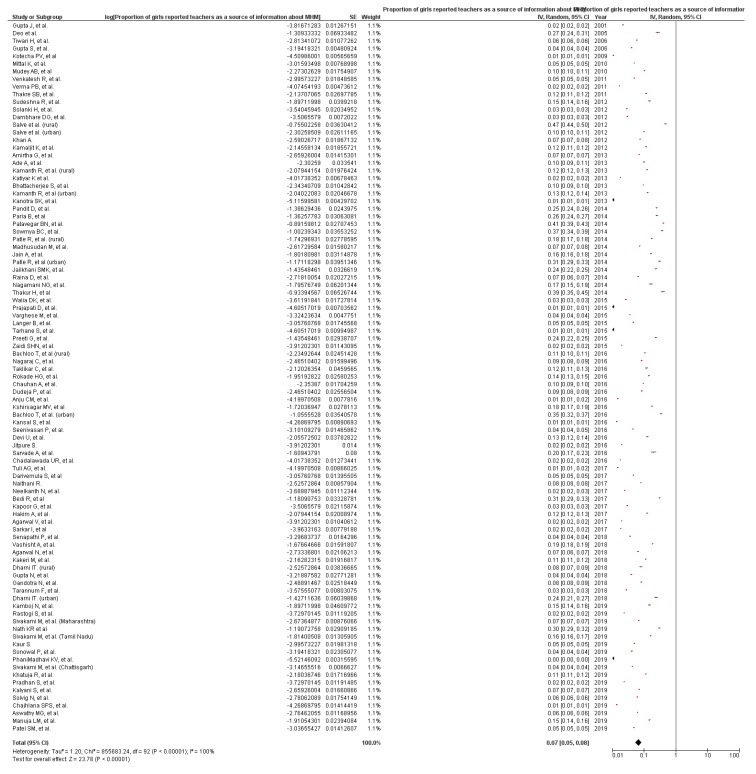
Pooled prevalence of teachers as a source of information about menstruation to girls in India, from the included studies published until October 2019 (n = 86 studies). CI: Confidence Interval; SE = Standard error. *I*^2^: Heterogeneity; Squares represent proportions or prevalence. Lines represent 95% CI. Diamonds represent pooled prevalence.

**Figure 4 ijerph-17-00647-f004:**
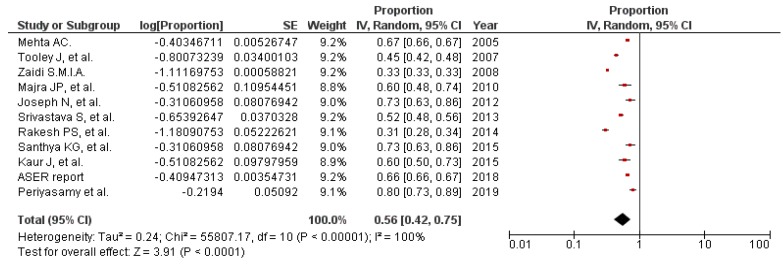
Pooled prevalence of schools with separate toilets for girls in India, from studies published until Oct 2019 (n = 11). CI: Confidence Interval; SE = Standard error; ASER: Annual Status of Education Report Centre. *I*^2^: Heterogeneity. Squares represent proportions or prevalence. Lines represent 95% CI. Diamonds represent pooled prevalence.

**Figure 5 ijerph-17-00647-f005:**
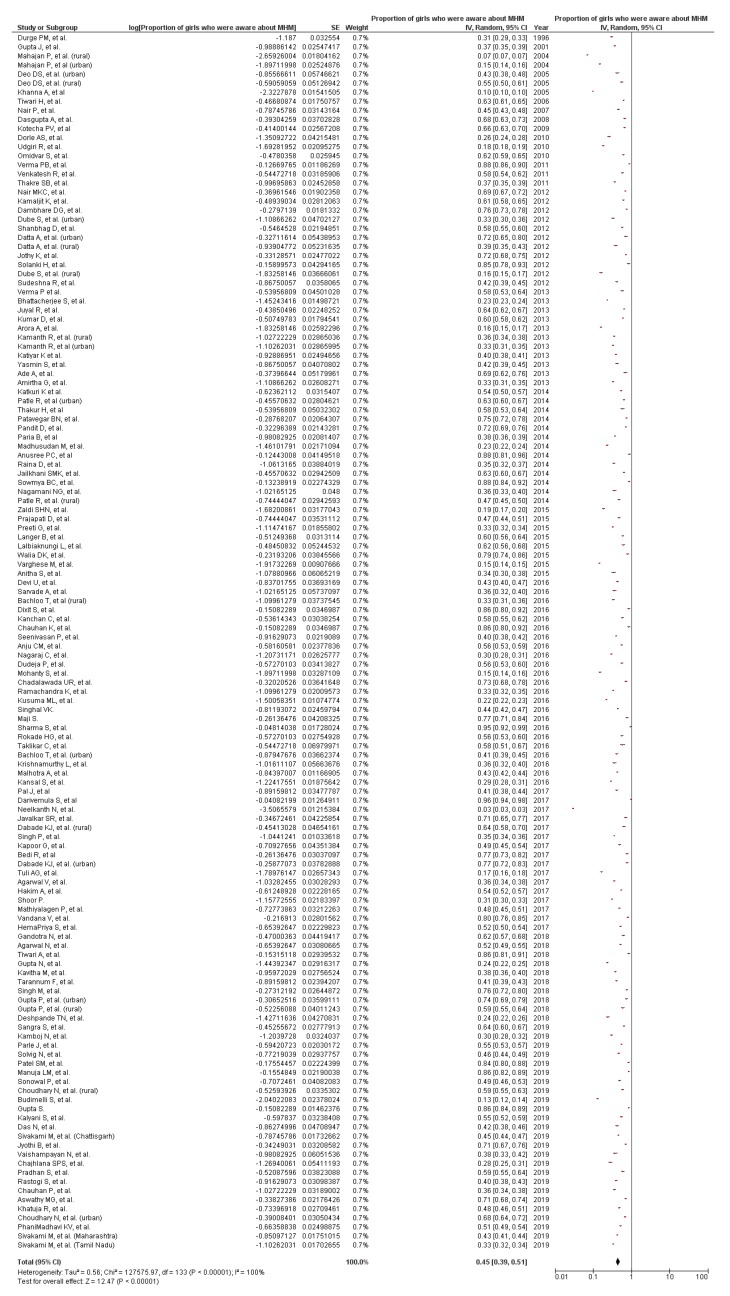
Pooled prevalence of girls’ awareness about menstruation prior to menarche in India, from studies published until Oct 2019 (n = 122). CI: Confidence Interval; SE = Standard error; *I*^2^: Heterogeneity. Squares represent proportions or prevalence. Lines represent 95% CI. Diamonds represent pooled prevalence

**Figure 6 ijerph-17-00647-f006:**
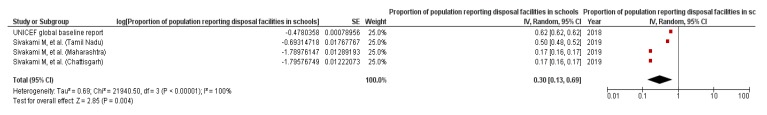
Pooled prevalence of proportion of girls/student populations reported that schools had good disposal facilities for sanitary products, from studies published until Oct 2019 (n = 2). CI: Confidence Interval; SE = Standard error; *I*^2^: Heterogeneity. Squares represent proportions or prevalence. Lines represent 95% CI. Diamonds represent pooled prevalence.

**Table 1 ijerph-17-00647-t001:** Characteristics of included studies related to menstrual hygiene friendly schools published until October 2019.

Serial Number	Variables	Number (%)
	School–Level Actions	N = 176
1	Type of publications	
	Original research	163 (92.6)
	Review articles	4 (2.3)
	Reports	9 (5.1)
2	Year of publication	
	1990–2005	8 (4.5)
	2006–2011	19 (10.8)
	2012–2017	103 (58.5)
	2018–2019	46 (26.2)
3	Study design of original research (n = 163)	
	Cross-section study	153 (94.0)
	Intervention study	10 (6.0)
4	Study population in original research (n = 163)	
	Girls	148 (90.7)
	Other populations only (teachers and boys)	3 (1.9)
	Schools	12 (7.4)
5	Settings in original research (n = 163)	
	Rural	62 (38.0)
	Urban and slums	42 (25.7)
	Both rural and urban	28 (17.2)
	Not clear from the study	31 (19.1)
6	Methods of data collection in original research (n =163)	
	Self-administered	67 (41.1)
	Questions asked by investigators	76 (46.6)
	Not clear from the study	20 (12.3)
7	Region (n = 163) ^¶^	
	North India	30 (18.4)
	South India	52 (31.9)
	East India	22 (13.5)
	West India	38 (23.4)
	Central India	19 (11.6)
	Mixed regions	2 (1.2)
8	Median sample size for (n = 160)	
	Adolescent girls (n = 148)	250
	Schools (n = 12)	28
	**Policy-level actions**	**N = 7**
1	Type of documents	
	Guidelines	4 (57.1)
	Reports	2 (28.5)
	Article	1 (14.4)
2	Year of publication	
	2014–2017	5 (71.4)
	2018–2019	2 (28.6)

^¶^**North**: New Delhi, Haryana, Jammu and Kashmir, Chandigarh, Himachal Pradesh, Punjab. **Central**: Chhattisgarh, Madhya Pradesh, Uttar Pradesh and Uttarakhand. **East**: Bihar, Jharkhand, Odisha, West Bengal, Meghalaya. **West:** Gujarat, Maharashtra, Rajasthan, Goa. **South:** Andhra Pradesh, Karnataka, Kerala, Tamil Nadu, Puducherry, Telangana, Hyderabad.

**Table 2 ijerph-17-00647-t002:** Characteristics of the included studies for teachers as a source of information about menstruation and girls’ pre-menarche awareness.

First Author	Year of Publication	Sample Size ^†^	Sample Size ^‡^	Location of the Study	Type of Area	Data Collection Method	Percent Awareness *	Percent Teachers as a Source **
Durge et al.	1996	200	-	Nagpur, Maharashtra	Urban	Unclear	30.5	-
Gupta et al.	2001	360	134	Jaipur, Rajasthan	Unclear	Self-administered	37.2	2.2
Mahajan et al.	2004	400 (Rural: 200; Urban: 200)	-	Jammu	Rural and urban	Self-administered	Rural:7.0 Urban:15.0	-
Deo et al.	2005	168 (Rural: 74; Urban: 94)	Rural: 41 Urban: NA	Ambajogai, Maharashtra	Rural and urban	Unclear	Rural:55.4 Urban:42.5	27.0
Khanna et al.	2005	372	-	Ajmer, Rajasthan	Rural and urban	Interview	9.8	-
Gupta et al.	2006	-	1700	Gorakhpur, Uttar Pradesh	Urban	Self-administered	-	4.1
Tiwari et al.	2006	763	486	Anand, Gujarat	Unclear	Self-administered	62.7	6.0
Nair et al.	2007	251	-	Delhi	Rural	Interview	45.5	-
Dasgupta et al.	2008	160	-	Hooghly, West Bengal	Rural	Self-administered	67.5	-
Nemade et al.	2009	217	-	Maharashtra	Unclear	Unclear	100.0	-
Kotecha et al.	2009	340	340	Vadodra, Gujarat	Rural	Self-administered	66.1	1.1
Mittal et al.	2010	-	788	Rohtak, Haryana	Urban	Interview	-	4.9
Udgiri et al.	2010	342	-	Bijapur, Karnataka	Urban	Unclear	18.4	-
Mudey et al.	2010	-	300	Wardha, Maharashtra	Rural	Self-administered	-	10.3
Omidvar et al.	2010	350	-	South India	Urban	Self-administered	62.0	-
Dorle et al.	2010	108	-	Bagalkot, Karnataka	Unclear	Self-administered	25.9	-
Verma et al.	2011	745	745	Bhavnagar, Gujarat	Urban	Self-administered	88.1	1.7
Thakre et al.	2011	387	143	Nagpur, Maharashtra	Rural and urban	Interview	36.9	11.8
Venkatesh et al.	2011	240	139	Bangalore, Karnataka	Rural and urban slums	Interview	58.0	5.0
Dube et al.	2012	200 (Rural: 100; Urban: 100)	-	Jaipur, Rajasthan	Rural and urban	Self-administered	Rural:16.0 Urban: 33.0	-
Jothy et al.	2012	330	-	Cuddalore, Tamil Nadu	Rural	Interview	71.8	-
Kamaljit et al.	2012	300	300	Amritsar, Punjab	Urban	Interview	61.3	11.7
Shanbhag et al.	2012	506	-	Bangalore, Karnataka	Rural	Self-administered	57.9	-
Sudeshna et al.	2012	190	80	Hooghly, West Bengal	Rural	Self-administered	42.0	15.0
Datta et al.	2012	155 (Rural: 87; Urban: 68)	-	Howrah, West Bengal	Rural and urban	Self-administered	Rural: 39.1 Urban: 72.1	-
Khan.	2012	-	199	Bellur, Karnataka	Rural	Interview	-	7.5
Solanki et al.	2012	68	68	Bhavnagar, Gujarat	Unclear	Self-administered	85.3	2.9
Dambhare et al.	2012	561 (Rural: 390; Urban: 171)	561	Wardha, Maharashtra	Rural and urban	Self-administered	75.6	3.0
Salve et al.	2012	-	Rural: 189 Urban: 132	Auranganbad, Maharashtra	Rural and urban	Interview	-	Rural: 47.0 Urban: 10.0
Nair et al.	2012	590	-	Thiruvanantha-puram, Kerala	Rural and urban	Self-administered	69.1	-
Verma et al.	2013	120	-	Varanasi, Uttar Pradesh	Unclear	Interview	58.3	-
Yasmin et al.	2013	147	-	Kolkata, West Bengal	Urban	Self-administered	42.0	-
Ade et al.	2013	80	80	Raichur, Karnataka	Rural	Self-administered	68.8	<10.0
Bhattacherjee et al.	2013	798	798	Siliguri, West Bengal	Slums	Interview	23.4	9.6
Juyal et al.	2013	453	-	Dehradun, Uttarakhand	Rural and semi-urban	Interview	64.5	-
Kanotra et al.	2013	-	323	Maharashtra	Rural	Self-administered	-	0.6
Kumar et al.	2013	744	-	Chandigarh	Rural and urban	Interview	60.2	-
Kamanth et al.	2013	550 (Rural: 280; Urban: 270)	550 (Rural: 280; Urban: 270)	Udupi, Karnataka	Rural and urban	Self-administered	Rural: 35.8 Urban: 33.2	Rural: 12.5 Urban: 13.0
Amirtha et al.	2013	325	325	Puducherry	Urban	Self-administered	33.0	7.0
Arora et al.	2013	200	-	Ambala, Haryana	Rural	Self-administered	16.0	-
Katiyar et al.	2013	384	384	Meerut, Uttar Pradesh	Urban	Interview	39.5	1.8
Paria et al.	2014	541	203	Kolkata, West Bengal	Rural and urban	Self-administered	37.5	25.6
Katkuri et al.	2014	250	-	Hyderabad	Unclear	Self-administered	53.6	-
Nagamani et al.	2014	100	36	Visakhapatnam, Andhra Pradesh	Urban slums	Interview	36.0	16.6
Raina et al.	2014	150	150	Dehradun, Uttarakhand	Rural	Interview	34.6	6.6
Patle et al.	2014	583 (Rural: 288; urban: 295)	324 (Rural: 187; urban: 137)	Nagpur, Maharashtra	Rural and urban	Interview	Rural: 47.5 Urban: 63.4	Rural: 17.5 Urban: 31.0
Pandit et al.	2014	435	315	Hooghly, West Bengal	Rural	Interview	72.4	25.0
Thakur et al.	2014	96	56	Mumbai, Maharashtra	Urban	Interview	58.3	39.3
Patavegar et al.	2014	440	330	Pulpralhadpur, Delhi	Urban	Self-administered	75.0	41.0
Anusree et al.	2014	60	-	Mangalore, Karnataka	Unclear	Self-administered	88.3	-
Jailkhani et al.	2014	268	170	Maharashtra	Urban slums	Interview	63.4	23.8
Sowmya et al.	2014	210	184	Bangalore, Karnataka	Rural	Interview	87.6	36.7
Jain et al.	2014	-	142	Nanded, Maharashtra	Rural	Interview	-	16.5
Madhusudan et al.	2014	378	271	Hosakote, Karnataka	Rural	Self-administered	23.2	7.3
Lalbiaknungi et al.	2015	86	-	Bhatar, West Bengal	Rural	Interview	61.6	-
Prajapati et al.	2015	200	200	Kheda, Gujarat	Rural	Interview	47.5	1.0
Zaidi et al.	2015	150	150	Thiruporur, Tamil Nadu	Unclear	Interview	18.6	2.0
Langer et al.	2015	245	147	Jammu and Kashmir	Rural	Interview	59.9	4.7
Preeti et al.	2015	640	210	Hyderabad	Urban	Unclear	32.8	23.8
Anitha et al.	2015	61	-	Udupi, Karnataka	Rural	Unclear	34.0	-
Varghese et al.	2015	1522	1522	Chennai, Tamil Nadu	Semi-urban	Self-administered	14.7	3.6
Walia et al.	2015	111	88	Chandigarh and Himachal Pradesh	Rural and urban	Interview	79.3	2.7
Tarhane et al.	2015	-	100	Nagpur, Maharashtra	Unclear	Self-administered	-	1.0
Mohanty et al.	2016	118	-	Berhampur, Odisha	Urban slums	Self-administered	15.0	-
Chadalawada et al.	2016	150	109	Vijayawada, Andhra Pradesh	Rural	Self-administered	72.6	1.8
Kansal et al.	2016	590	174	Chiraigaon, Varanasi	Rural	Interview	29.4	1.4
Seenivasan et al.	2016	500	200	North Chennai, Tamil nadu	Urban	Unclear	40.0	4.5
Kshirsagar et al.	2016	-	190	Maharashtra	Rural	Interview	-	17.9
Kanchan et al.	2016	263	-	Hyderabad	Rural and urban	Self-administered	58.5	-
Dudeja et al.	2016	211	119	Maharashtra	Urban slum	Self-administered	56.4	8.5
Anju et al.	2016	436	244	Perinthalmanna, Kerala	Rural	Self-administered	55.9	1.5
Ramachandra et al.	2016	550	-	Bangalore, Karnataka	Urban	Self-administered	33.3	-
Devi et al.	2016	180	78	Kancheepuram, Tamil Nadu	Rural	Self-administered	43.3	12.8
Taklikar et al.	2016	50	50	Kolkata, West Bengal	Urban slum	Interview	58.0	12.0
Nagaraj et al.	2016	304	304	Bangalore, Karnataka	Rural	Self-administered	29.9	8.5
Maji.	2016	100	-	Burdwan, West Bengal	Rural	Interview	77.0	-
Chauhan et al.	2016	-	296	Ahmadabad	Urban	Interview	-	9.5
Malhotra et al.	2016	1800	-	Uttar Pradesh	Rural	Interview	43.0	-
Krishnamurthy et al.	2016	72	-	Kolar, Karnataka	Rural	Self-administered	36.2	-
Singhal.	2016	408	-	Gurgaon, Haryana	Rural	Interview	44.4	-
Sarvade et al.	2016	70	25	Mumbai, Maharashtra	Urban	Self-administered	36.0	20.0
Sharma et al.	2016	150	-	Aroha, Haryana	Rural	Self-administered	95.3	-
Kusuma et al.	2016	1500	-	Mysore, Karnataka	Urban	Interview	22.3	-
Rokade et al.	2016	324	183	Solapur, Maharashtra	Urban	Interview	56.4	14.2
Bachloo et al.	2016	Rural: 159 Urban: 181	Rural: 159 Urban: 181	Ambala, Haryana	Rural and urban	Self-administered	Rural: 33.3 Urban: 41.5	Rural: 10.7 Urban: 34.8
Jitpure.	2016	-	100	Bilaspur, Chhattisgarh	Urban slums	Self-administered	-	2.0
Chauhan et al.	2016	100	-	Shimla, Himachal Pradesh	Rural	Interview	86.0	-
Dixit et al.	2016	100	-	Indore, Madhya Pradesh	Urban	Self-administered	86.0	-
Pal et al.	2017	200	-	Kolkata, West Bengal	Urban slum	Self-administered	41.0	-
Darivemula et al.	2017	240	230	Andhra Pradesh	Rural	Interview	96.0	4.7
Sarkar et al.	2017	-	307	Hooghly, West Bengal	Rural	Self-administered	-	1.9
Bedi et al.	2017	192	192	Ajmer, Rajasthan	Rural	Self-administered	77.0	30.7
Javalkar et al.	2017	116	-	Mangalore, Karnataka	Rural	Interview	70.7	-
Tuli et al.	2017	197	197	Ludhiana, Punjab	Rural	Interview	16.7	1.5
Singh et al.	2017	2135	-	Udham Singh Nagar, Uttarakhand	Unclear	Interview	35.2	-
Hakim et al.	2017	500	271	Jodhpur, Rajasthan	Urban	Interview	54.2	12.5
Agarwal et al.	2017	250	181	Sabarkantha, Gujarat	Rural	Interview	35.6	2.0
Mathiyalagen et al.	2017	242	-	Puducherry	Rural and urban	Interview	48.3	-
Shoor.	2017	452	-	Tumkur, Karnataka	Urban	Interview	31.4	-
Neelkanth et al.	2017	197	197	Bhopal, Madhya Pradesh	Unclear	Self-administered	3.0	2.5
Naithani.	2017	-	1000	Pauri, Uttarakhand	Rural and urban	Interview	-	8.0
Vandana et al.	2017	200	-	Ambala, Haryana	Rural	Unclear	80.5	-
Dabade et al.	2017	Rural: 107 Urban: 123	-	Gulbarga, Karnataka	Rural and urban	Unclear	Rural: 63.5 Urban: 77.2	-
Kapoor et al.	2017	132	65	Jammu	Rural	Interview	49.2	3.0
Hemapriya et al.	2017	502	-	Puducherry	Rural	Interview	52.0	-
Gandotra et al.	2018	120	120	Dehradun, Uttarakhand	Urban	Unclear	62.5	8.3
Deshpande et al.	2018	100	-	Karad, Maharashtra	Urban slum	Interview	24.0	-
Gupta et al.	2018	212	50	Etawah, Uttar Pradesh	Rural	Interview	23.6	4.0
Agarwal et al.	2018	263	137	Raipur, Chhattisgarh	Rural	Interview	52.0	6.5
Tarannum et al.	2018	422	422	Aligarh, Uttar Pradesh	Unclear	Unclear	41.0	2.8
Vashisht et al.	2018	-	600	Delhi	Unclear	Interview	-	18.7
Tiwari et al.	2018	141	-	Rajnandgaon, Chhattisgarh	Rural	Interview	85.8	-
Kakeri et al.	2018	-	277	Palghar, Maharashtra	Rural	Self-administered	-	11.5
Dharni.	2018	-	Rural: 50 Urban: 50	Ludhiana, Punjab	Rural and urban	Interview	-	Rural: 8.0 Urban: 24.0
Kavitha et al.	2018	311	-	Bangalore, Karnataka	Rural	Self-administered	38.3	-
Senapathi et al.	2018	-	132	Mangaluru, Karnataka	Rural	Interview	-	3.7
Gupta et al.	2018	Rural =150; Urban =150	-	Kota, Rajasthan	Rural and urban	Self-administered	Rural: 59.3 Urban: 73.6	-
Singh et al.	2018	260	-	Gurugram, Haryana	Unclear	Self-administered	76.1	-
Chauhan et al.	2019	226	-	South India	Rural	Self-administered	35.8	-
Aswathy et al.	2019	432	432	Thrissur, Kerala	Unclear	Self-administered	71.3	6.3
Rastogi et al.	2019	250	187	Delhi	Unclear	Interview	40.0	2.4
Sivakami et al.	2019	C* = 826 M* = 798 T* = 765	C* = 927 M* = 837 T* = 800	Chhattisgarh, Maharashtra, Tamil Nadu	Unclear	Self-administered	C* = 45.5 M* = 42.7 T* = 33.2	C* = 4.3 M* = 6.9 T* = 16.3
Patel et al.	2019	273	229	Mandur, Goa	Rural	Self-administered	83.9	4.8
Nath et al.	2019	-	250	Kanyakumari district Tamil Nadu	Rural	Interview	-	30.4
Khatuja et al.	2019	340	340	Delhi	Urban slums	Interview	48.0	11.3
Pradhan et al.	2019	165	165	Cuttak, Odisha	Urban	Unclear	59.4	2.4
Sangra et al.	2019	300	-	Kathua, Jammu and Kashmir	Rural	Self-administered	63.6	-
Madhavi et al.	2019	400	400	Andhra Pradesh	Rural	Interview	51.5	0.4
Chajhlana et al.	2019	69	69	Hyderabad	Urban	Unclear	28.1	1.4
Budimelli et al.	2019	200	-	Guntur, Andhra Pradesh	Rural	Interview	13.0	-
Das et al.	2019	110	-	Jorhat, Assam	Urban slums	Interview	42.2	-
Sonowal et al.	2019	150	74	Dibrugarh, Assam	Urban slums	Interview	49.3	4.1
Jyothi et al.	2019	200	-	Bagalkot, Haryana	Urban	Interview	71.0	-
Choudhary et al.	2019	Rural: 215 Urban: 235	-	Jodhpur, Rajasthan	Rural and urban	Self-administered	Rural: 59.1 Urban: 67.7	-
Kamboj et al.	2019	200	60	Sirsa, Haryana	Rural	Self-administered	30.0	15.0
Gupta.	2019	563	-	Sambhal, Uttar Pradesh	Rural	Unclear	86.0	-
Parle et al.	2019	600	-	Raigad, Maharashtra	Rural	Self-administered	55.2	-
Kaur.	2019	-	121	Kapurthala, Punjab	Unclear	Unclear	-	5.0
Solvig et al.	2019	288	189	Bangalore, Karnataka	Rural and urban	Interview	46.2	6.2
Manuja et al.	2019	257	220	Mandya, Karnataka	Rural	Self-administered	85.6	14.8
Vaishampayan et al.	2019	64	-	Telangana	Unclear	Self-administered	37.5	-
Kalyani et al.	2019	236	236	Goa	Unclear	Self-administered	55.0	7.0

* Percent girls aware about menstruation prior to menarche. ** Percent girls reported teachers as a source of information on menstruation. ^†^ Sample size for MHM awareness and ^‡^ Sample size for teachers as a source of information; C*= Chhattisgarh; M*= Maharashtra; T*= Tamil Nadu.

**Table 3 ijerph-17-00647-t003:** Characteristics of the included studies for separate toilet for girls and good disposal facilities for sanitary products.

First Author	Year of Publication	Sample size	Location of the Study	Type of Area	Data Collection Method	Percent Schools with Disposal Facilities for Sanitary Products ^β^	Percent Schools have Separate Toilets for Girls
Sivakami et al.	2019	C*:927 girls M*:837 girls T*:800 girls (total: 53 schools)	Chhattisgarh Maharashtra Tamil Nadu	Unclear	Self-administered	C*:16.6 M*:16.7 T*:50.0 (average: 27.1)	C*:29.4 M*:20.3 T*:61.3 (average: 36.5)
UNICEF global baseline report.	2018	377,929 school age population	India	Rural and urban	Secondary data	62.0	73.0
Periyasamy et al.	2019	61 schools	Bengaluru	Urban	Interview	NA	80.3
Srivastava et al.	2013	182 schools	Uttar Pradesh	Rural and urban	Interview	51.0	52.0
ASER report	2018	17,730 schools	India	Rural	Survey	NA	66.4
Mehta.	2005	7993 schools	Punjab	Rural and urban	Secondary data	NA	66.8
Tooley et al.	2007	214 schools	East Delhi	Slums	Census and survey	NA	44.9
Zaidi S.M.I.A.	2008	638,057 schools	India	Rural and urban	Secondary data	NA	32.9
Majra et al.	2010	20 schools	Karnataka	Rural	Interview	NA	60.0
Joseph et al.	2012	30 schools	Karnataka	Urban	Interview	NA	73.3
Rakesh et al.	2014	78 schools	Kollam, Kerala	Unclear	Interview	NA	30.7
Kaur et al.	2015	25 schools	Chandigarh	Rural and urban	Interview	NA	60.0
Santhya et al.	2015	30 schools	Bihar	Rural and urban	Self-administered and interview	NA	73.3

C*: Chhattisgarh; M*: Maharashtra; T*: Tamil Nadu. ^β^ In Sivakami et al. study, it is the proportion of girls reporting on the presence of disposal facilities in schools. Abbreviations: ASER: Annual Status of Education Report Centre; NA: Not Available.

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
