# Peer review of "Menstrual Hygiene Preparedness Among Schools in India: A Systematic Review and Meta-Analysis of System-and Policy-Level Actions"

_ijerph, 2020, doi:10.3390/ijerph17020647_

Round 1

Reviewer 1 Report

Sharma et al., tried to summarize the evidence on the actions taken at the school (system)- and policy-level to make schools a menstrual hygiene friendly place for adolescent girls in India by systematically reviewing the papers published until Oct 2019 . The following major concerns should be addressed.

1.      This paper is statistically very poor and furthermore analysis should be done for better understanding (meta-regression analysis)

2.       It is better to differentiate rural and urban for comparison purpose.

3.      There are many issues to be identified and their correlations should be analyzed

4.       The abstract and discussion is too general and it should be improved.

5.      PRISMA framework checklist should be submitted as supplementary file

Author Response

Sharma et al., tried to summarize the evidence on the actions taken at the school (system)- and policy-level to make schools a menstrual hygiene friendly place for adolescent girls in India by systematically reviewing the papers published until Oct 2019. The following major concerns should be addressed.

Comment 1: This paper is statistically very poor and furthermore analysis should be done for better understanding (meta-regression analysis)

Reply: We have now done meta-analysis on 4 out of the eight components for which we had data.

Comment 2: It is better to differentiate rural and urban for comparison purpose.

Reply: We tried to do it, but not much difference was found in meta-analysis.

Comment 3: There are many issues to be identified and their correlations should be analyzed.

Reply: We have looked at only 8 components of the menstrual hygiene friendly schools.

Comment 4: The abstract and discussion is too general and it should be improved.

Reply: We have now improved the abstract and discussion to make it more specific.

Comment 5: PRISMA framework checklist should be submitted as supplementary file.

Reply: We will add the PRISMA checklist as a supplementary file.

Reviewer 2 Report

How were the eight criteria decided? Sorry if I missed these details.

285 The government launched 100 percent oxy-biodegradable sanitary napkins under the name “Suvidha (facility)” to be available at Pradhan Mantri Bhartiya Janaushadhi Pariyojana (Prime Minister Indian People Drug
 Scheme) [75]. -- this last piece is confusing and needs more information

The first paragraph of the discussion needs to be more clear; remember some people will skip the results and just read this.

two components, primarily girls’ awareness about MHM, and sanitation facilities in schools, leaving other components unaddressed. MHM in schools, although it was conceptualized comprehensively with different components as documented in guidelines, the data on its implementation was limited. It is imperative to emphasize  the four primary considerations to build effective evidence on MHM friendly school aspect.

OR even just saying -- these components are described further, below.

313. Regular monitoring and timely actions are crucial to transform the current scenario of schools into MHM friendly. (This sentence needs to be edited for grammar). 

Author Response

Comment 1: How were the eight criteria decided? Sorry if I missed these details.

Reply: The eight criteria were based on a previously published report. The reference for the same is:

Kirk, J.; Sommer, M. Menstruation and body awareness: linking girls’ health with girls’ education. In Special on Gender and Health; Royal Tropical Institute (KIT): Amsterdam, The Netherlands, 2006; pp. 1-22.

Comment 2: 285 The government launched 100 percent oxy-biodegradable sanitary napkins under the name “Suvidha (facility)” to be available at Pradhan Mantri Bhartiya Janaushadhi Pariyojana (Prime Minister Indian People Drug Scheme) [75]. -- this last piece is confusing and needs more information

Reply:  I have now added a few lines on this issue to make it more clear.

The government launched 100 percent oxy-biodegradable sanitary napkins under the name “Suvidha (facility)”. These sanitary pads are available under the scheme, entitled “Pradhan Mantri Bhartiya Janaushadhi Pariyojana” (Prime Minister Indian People Drug Scheme) [75]. The sanitary pads were made available at INR 1 in the drug dispensing stores created under the scheme. These napkins biodegrade automatically when it comes in contact with oxygen after being discarded [75].

Comment 3: The first paragraph of the discussion needs to be more clear; remember some people will skip the results and just read this.

Reply: yes, I have now made it more specific.

Comment 4: two components, primarily girls’ awareness about MHM, and sanitation facilities in schools, leaving other components unaddressed. MHM in schools, although it was conceptualized comprehensively with different components as documented in guidelines, the data on its implementation was limited. It is imperative to emphasize  the four primary considerations to build effective evidence on MHM friendly school aspect.

OR even just saying -- these components are described further, below.

Reply: I have tried to change it.

Comment 5: 313. Regular monitoring and timely actions are crucial to transform the current scenario of schools into MHM friendly. (This sentence needs to be edited for grammar).

Reply: Regular monitoring and timely actions are crucial to transform the poor MHM practices in schools.

Reviewer 3 Report

I read with great interest the Manuscript titled “Menstrual hygiene preparedness among schools in India: A scoping review of system-and policy-level actions” (ijerph-655106), which falls within the aim of International Journal of Environmental Research and Public Health.    

In my honest opinion, the topic is interesting enough to attract the readers’ attention. Nevertheless, authors should clarify some methodological points and improve the discussion citing relevant and novel key articles about the topic.

Authors should consider the following recommendations:

Manuscript should be further revised by a native English speaker. Inclusion/exclusion criteria of the studies should be better clarified. On the basis of what criteria did the authors exclude the studies from their analysis? The Authors did not report the results of the assessment of risk of bias. I think this is very important information to report so I suggest that the authors include these results in their manuscript. It is possible to report the risk of bias assessment as table. Was this review registered in PROSPERO? I could not find any information about this point. I would stress the significant potential impact of dysmenorrhea and pelvic pain of psychological wellbeing in young women affected by endometriosis (refer to: PMID: 28553145; PMID: 27750472; PMID: 27986180; J Endometr Pelvic Pain Disord. 2017;9(4):270-274. DOI: 10.5301/jeppd.5000303.), and the necessity to adopt proper strategies to raise awareness especially in schools.

Author Response

Reply: We have tried to edit it to some extent.

Comment 2: Inclusion/exclusion criteria of the studies should be better clarified. On the basis of what criteria did the authors exclude the studies from their analysis?

Reply: Although I mentioned in the methodology section that the studies, which did not report on any one of the eight criteria for menstrual hygiene friendly schools were excluded from the analysis. Also, I added another line:

“Studies that have included girl’s awareness on MHM in their findings, but did not mention that they were school going girls were also excluded from the analysis”

Comment 3: The Authors did not report the results of the assessment of risk of bias. I think this is very important information to report so I suggest that the authors include these results in their manuscript. It is possible to report the risk of bias assessment as table.

Reply: We have done the quality assessment of the studies, but we did not exclude studies based on the scores.

Comment 4: Was this review registered in PROSPERO? I could not find any information about this point.

Reply: We could not this earlier since our objective was to do a scoping review. However, this platform is aimed for systematic reviews, and not scoping reviews, and that to our knowledge, no register exists specific for scoping reviews.

Comment 5: I would stress the significant potential impact of dysmenorrhea and pelvic pain of psychological wellbeing in young women affected by endometriosis (refer to: PMID: 28553145; PMID: 27750472; PMID: 27986180; J Endometr Pelvic Pain Disord. 2017;9(4):270-274. DOI: 10.5301/jeppd.5000303.), and the necessity to adopt proper strategies to raise awareness especially in schools.

Reply: I have now added this component in the discussion section.

“Besides, menstrual irregularities during reproductive age group are common in many gynaecological diseases, such as endometriosis, which may affect mental and psychological well-being in long-term [84].”.

Round 2

Reviewer 1 Report

Sharma et al tried to address all the comments raised.